# Analysis of Antioxidant and Antiviral Effects of Olive (*Olea europaea* L.) Leaf Extracts and Pure Compound Using Cancer Cell Model

**DOI:** 10.3390/biom13020238

**Published:** 2023-01-27

**Authors:** Rosamaria Pennisi, Ichrak Ben Amor, Bochra Gargouri, Hamadi Attia, Rihab Zaabi, Ahlem Ben Chira, Mongi Saoudi, Anna Piperno, Paola Trischitta, Maria Pia Tamburello, Maria Teresa Sciortino

**Affiliations:** 1Department of Chemical, Biological, Pharmaceutical, and Environmental Sciences, University of Messina, Viale SS. Annunziata, 98168 Messina, Italy; 2Unit of Biotechnology and Pathologies, Higher Institute of Biotechnology of Sfax, University of Sfax, Sfax 3029, Tunisia; 3Animal Ecophysiology Laboratory, Sciences Faculty of Sfax, University of Sfax, Sfax 3054, Tunisia

**Keywords:** *Olea europaea* L., oleuropein, oxidative stress, antiviral activity, herpes simplex virus 1, innate immune response, PKR

## Abstract

The present study aims to assess the antioxidant and antiviral effectiveness of leaf extracts obtained from *Olea europaea* L. var. *sativa* and *Olea europaea* L. var. *sylvestris*. The total antioxidant activity was determined via both an ammonium phosphomolybdate assay and a nitric oxide radical inhibition assay. Both extracts showed reducing abilities in an in vitro system and in human HeLa cells. Indeed, after oxidative stress induction, we found that exposition to olive leaf extracts protects human HeLa cells from lipid peroxidation and increases the concentration of enzyme antioxidants such as catalase (CAT), superoxide dismutase (SOD), and glutathione peroxidase. Additionally, OESA treatment affects viral DNA accumulation more than OESY, probably due to the exclusive oleuropein content. In fact, subtoxic concentrations of oleuropein inhibit HSV-1 replication, stimulating the phosphorylation of PKR, c-FOS, and c-JUN proteins. These results provide new knowledge about the potential health benefits and mechanisms of action of oleuropein and oleuropein-rich extracts.

## 1. Introduction

The use of plants as a source of medicinal herbs has increased tremendously over the past three decades because of important advantages such as safety, efficacy and global availability [1]. Among these, the medicinal plants of the Mediterranean area are well known for their biological properties in traditional medicinal, gastronomic, and industrial use. Additionally, they have played significant roles in drug discovery and development, especially against cancer [2]. Olive leaves are widely used in the human diet due to their impact on human health as antioxidants, hypoglycemics, antihypertensives, antimicrobials, antivirals, and antiatherosclerotics [3]. This property can be related to their phenolic content. It was reported that olive leaves contain high amounts of secoiridoids, flavonoids, and simple phenols, such as *oleuropein*, *luteolin*, *luteolin-7-O-glucoside*, and *hydroxytyrosol* [4]. Polyphenols modulate oxidative stress in cancer cells through the modulation of signal transduction and the expression of specific genes related to cell proliferation and cell death [5,6]. Polyphenol compounds trigger apoptotic programmed cell death pathways in human gastric carcinoma cells by manipulating the reactive oxygen species (ROS) [5]. ROS can react with biological molecules, such as DNA, proteins, or lipids, generating mutations and damaging membranes, leading to cell and tissue injuries [7,8,9].

Among the antioxidant enzymes, catalase (CAT), superoxide dismutase (SOD), and glutathione peroxidase (GPx) play key roles [10]. Thus, the intake of exogenous antioxidants from fruits or vegetables can support the antioxidant defense system and reduce oxidative stress and the incidence of certain types of cancer in which ROS accumulation overwhelms scavenging activity [10]. The olive leaves from *O. europaea* contain a mixture of polyphenolic compounds [11,12] which exhibit a broad spectrum of biological activities such as antimicrobial, antiviral, anti-inflammatory, anti-carcinogenic, anti-allergic, anti-thrombotic, cardioprotective, and vasodilatory effects [4,13,14,15,16].

In this regard, we first evaluated the hydroethanolic extracts of two varieties of *Olea europaea* L. as a new potential source of natural antioxidants using a cancer cell model: the human cervical adenocarcinoma HeLa cell line. In fact, oxidants and free radicals induce DNA damage and represent the triggering factors for cancer pathogenesis. Thus, the anti-free radical activity of natural compounds can be useful in chemoprevention and chemotherapy. Therefore, we assessed the biological activity of olive leaf extracts from the cultivar (*O. europaea* L. ssp. *europaea* var. *sativa*) and the true wild oleaster (*O*. *europaea* L. var. *sylvestris*). The antioxidant activity was chemically determined via an in vitro assay (phosphomolybdenum assay) and scavenging activity of nitric oxide (NO), and was biologically assessed using the HeLa cell line. Because cancers that occur secondarily to infections are the most common diseases and viral infection based on herpes simplex virus type I is considered one of the cofactors, next, we investigated the anti-HSV-1 activity of the OESA and OESY olive leaf extracts. We also investigated the antiviral role of oleuropein, which is among the most abundant phytochemical compounds exclusively reported (Figure 1) in OESA olive leaf extracts, and we used a cancer cell as a cellular model. The outcomes of this study will provide scientific evidence to support the efficacy and the use of *Olea europaea* L. and its bioactive compounds as antioxidant adjuvants and antivirals in HeLa cancer cells.

## 2. Materials and Methods

### 2.1. Plant Materials and Extraction Procedure

The samples of the olive leaves extracted from *O. europaea* var. *sativa* (OESA) and *O. europaea* var. *sylvestris* (OESY) were produced from the variety Chemlali SFAX. Olive leaves were collected manually from the National agricultural complex “Chaal” located in the Sfax region (latitude: 34°44′26′′ north; longitude: 10°45′37′′ east), Tunisia. Samples were harvested in the same pedoclimatic conditions during the month of October 2018. The climate in this area is arid and semiarid with irregular and torrential precipitation [17]. Olive leaves were dried at room temperature, and 100 g of plant leaf material was treated overnight with water: ethanol 50:50 (*v*/*v*) under gentile stirring. The hydroethanolic extract was filtered through a cellulose filter, lyophilized, and frozen at −80 °C until use. The extracts were previously characterized via using LC-DAD-ESI-MS analysis (Phenomenex, CA, USA) [16]. Oleuropein was obtained from fresh olive leaves (*Olea europaea* L.). In brief, 50 g of fresh olive leaves (*Olea europaea* L.) in 250 mL of water were boiled for 2 h. The solution was recovered and evaporated under a vacuum. The resulting residue was extracted several times with hot acetone, and then, the organic solutions were collected, and after 12 h, filtered and evaporated under a vacuum. The residue was purified via treatment with hot acetone twice to yield 150 mg of oleuropein, which was identified by comparing the ^1^H and ^13^C NMR spectra with the literature data [18,19].

### 2.2. Antioxidant Capacity Assays

#### 2.2.1. Phosphomolybdenum Assay

Extract samples were mixed with 1 mL of the phosphomolybdenum reagent (600 mM sulfuric acid, 4 mM ammonium molybdate, and 28 mM sodium phosphate) [20,21]. Then, the mixture was incubated at 95 °C for 90 min and cooled to room temperature. The absorbance was measured at 695 nm. A standard curve was constructed using ascorbic acid to estimate the percentage of molybdenum reduced by the tested extracts. EC50 (mg/mL) corresponds to the effective concentration at which the total antioxidant activity (TAA) at 50% was obtained via interpolation from the linear regression analysis.

#### 2.2.2. Scavenging Activity of Nitric Oxide (NO)

NO scavenging activity of extracts was determined as previously described [22]. Briefly, various concentrations of extracts (0 to 1 mg/mL) and Vitamin C were incubated with 0.5 mL of sodium nitrite (0.01 mg/mL in 100 mM sodium citrate, pH 5) at 37 °C for 2 h. After incubation, 0.5 mL of Griess reagent was added, and the absorbance (A) was measured at 540 nm using a spectrophotometer Bichrom Libra S32 (Beckman, Fullerton, CA, USA). The Equation (1) obtained the percentage of reactive nitrogen species (RNS) scavenging:

NO scavenging effect (%) = (A0 − A1)/A0 × 100 (1)
where A0 is the absorbance of the control and A1 is the absorbance of the sample.

### 2.3. HeLa Cell Culture

Continuous human HeLa (epithelial cervical cancer cell line) cell lines, supplied by ATCC (Manassas, VA, USA), were employed to test the cytotoxicity and antioxidant effects of plant extracts [15]. The cells were grown in RPMI 1640 medium (Gibco, Grand Island, NY, USA) supplemented with 10% (*v/v*) fetal calf serum (FCS) and 2 mM L-glutamine. VERO cell lines (American Type Culture Collection) were propagated in minimal essential medium (EMEM) and supplemented with 6% fetal bovine serum (FBS) (Lonza, Belgium). Both cells were kept at 37 °C in a humidified atmosphere of 95% air and 5% CO_2_.

### 2.4. Cell Proliferation Assay

To assess the HeLa cells’ viability following treatment with OESA and OESY, a CCK-8 assay (ab228554; Abcam, Cambridge, UK) was performed based on the manufacturer’s instructions. WST-8/CCK8 tetrazolium salt is reduced by cellular dehydrogenases to form an orange formazan product that is soluble in tissue culture medium. The amount of formazan produced is directly proportional to the number of living and metabolically active cells and is measured via absorbance at 460 nm. Therefore, HeLa cells (2 × 10^4^ cells/mL) were grown in 96-well microtiter plates at 37 °C in a 5% CO_2_ incubator for 24 h. Then, they were exposed to serial dilutions of OESA and OESY (0.1; 0.2; 0.4; 1; 5; 10; and 20 mg/mL) and oleuropein (25, 50, 100, 150, 200, 300 and 400 µg/mL) for 72 h and incubated with CCK8 tetrazolium salt for 4 h at 37 °C in a CO_2_ incubator. The absorbance was measured at 460 nm using a GloMax^®^ Discover Microplate Reader (Promega, Madison, Wisconsin, United States) and the percentage of cellular viability was calculated compared to untreated cells.

### 2.5. Induction of Oxidative Stress

For the induction of oxidative stress, HeLa cells (3 × 106 cells/mL in 25 cm^2^ flasks) were incubated at 37 °C for 72 h and later exposed to 0.2 mM of H_2_O_2_ for 1 h. After treatment with H_2_O_2_, the cells were rinsed three times with PBS, trypsinized, and centrifuged at 3000 rpm for 10 min. Finally, the cell pellet was recovered and used for further experiments.

### 2.6. Malondialdehyde (MDA) Determination

For evaluation of the MDA production rate, a thiobarbituric acid-reactive species (TBARs) assay was used as reported in a previous study [15,23]. Briefly, the pellet was resuspended in 500 μL of deionized water and lysed via five cycles of sonication for 20 s at 35% (Sonics, Vibra-Cell, Newtown, CT, USA). One milliliter of TBA solution (15% trichloroacetic acid, 0.8% thiobarbituric acid, and 0.25 N of HCl) was added. The mixture was heated at 95 °C for 15 min to form an MDA-TBA adduct. Optical density (OD) was measured using a spectrophotometer (Biochrom, Libra S32 Waterbeach, Cambridge, UK) at 532 nm. Values were reported to a calibration curve of 1,1,3,3-tetraethoxypropane (1.1.3.3 TEP).

### 2.7. Antioxidant Effect

To assay the capacity of plant extracts to protect HeLa cells from ROS-mediated oxidative injury, cells were exposed for 72 h to different concentrations (0.4 and 0.2 mg/mL) of ethanolic extracts (OESA and OESY), and then, treated with an oxidative stress-inducing agent (0.2 mM of H_2_O_2_) for 30 min.

### 2.8. Measurement of Antioxidant Enzyme Activity

#### 2.8.1. Determination of Catalase Activity

Catalase activity (CAT) was measured as described previously by Aebi [24]. A total of 200 µL of cell lysate and 1790 µL of potassium phosphate buffer (50 mM, pH 7.0) were combined in a UV cuvette to evaluate the catalase activity. A total of 10 µL of H_2_O_2_ (30%) was added, and the catalase-mediated elimination of H_2_O_2_ was monitored at 240 nm. Under standard conditions, the amount of H_2_O_2_ converted into H_2_ and O_2_ in 1 min was accepted as the enzyme reaction velocity. The enzyme activity was calculated using an extinction coefficient of 0.043 mM^−1^.cm^−1^ and expressed as µmol H_2_O_2_ destroyed/min/mg (µmol/minute/mg protein).

#### 2.8.2. Superoxide Dismutase Activity Assay

The superoxide dismutase (SOD) activity was estimated according to the colorimetric method of Beyer and Fridovich [25]. Cell lysate mixed in 0.1 M of potassium phosphate buffer (pH 7.4), 0.1 mM EDTA, 13 mL methionine, 2 mM riboflavin, and 75 mM nitro-blue tetrazolium (NBT). The developed blue color in the reaction was measured at 560 nm. Units of SOD activity were expressed as the amount of enzyme required to inhibit the reduction of NBT by 50%. The activity was expressed as units per mg of protein.

#### 2.8.3. Glutathione Peroxidase Activity Assay

Glutathione peroxidase (GPx) activity was measured according to the procedure of Flohe and Gunzler [26]. The supernatant obtained after centrifuging 5% cells at 1500× *g* for 10 min, followed by 10,000× *g* for 30 min at 4 °C, was used for the GPx assay. A total of 1 mL of the reaction mixture, containing 0.3 mL of phosphate buffer (0.1 M, pH 7.4), 0.2 mL of GSH (2 mM), 0.1 mL of sodium azide (10 mM), 0.1 mL of H_2_O_2_ (1 mM), was added to 0.3 mL of cells supernatants. After incubation at 37 °C for 15 min, the reaction was terminated by adding 0.5 mL 5% TCA. Tubes were centrifuged at 1500× *g* for 5 min. The supernatant was collected in 0.2 mL of phosphate buffer (0.1 M pH 7.4), and 0.7 mL of DTNB (0.4 mg/mL) was added to 0.1 mL of the reaction supernatant. After mixing, absorbance was recorded at 420 nm, and the enzyme activity was calculated as micromoles of GSH/min/mg protein.

### 2.9. Standard Plaque Assay on VERO Cells

Confluent monolayers of VERO cells were prepared in 12 multiwell plates. The standard plaque assay procedure was previously reported [27]. The infected samples were frozen and thawed three times and diluted. A total of 100 µL of each dilution of the suspension was used to infect the monolayers. The multiwell plates were incubated for 1 h at 37 °C. Then, the viral inoculum was removed and 1ml of culture medium containing 0.8% methylcellulose was added. After 72 h, the plaques were visualized and counted under a microscope after staining with a crystal violet solution.

### 2.10. Viral DNA Extraction and Real-Time PCR

The viral DNA was extracted using phenol/chloroform solution and precipitated from the organic phase. The procedures were published previously in [28]. The DNA pellet was washed twice in a solution containing 0.1 M trisodium citrate in 10% ethanol, and then, dissolved in 8 mM NaOH. The concentration of DNA was determined via fluorometer analysis using a Qubit double-stranded DNA (dsDNA) HS (High Sensitivity) Assay Kit according to the manufacturer’s instructions. The amplification of viral DNA was carried out using TaqMan™ Universal Master Mix II (Applied Biosystems™, Foster City, CA, USA) in a 50 µL reaction mixture containing: TaqMan Universal Master Mix II, DNA (100 ng), HSV-1 forward (10 µM) and reverse (10 µM) primers (Fw 5′-catcaccgacccggagagggac; Rev 5′-gggccaggcgcttgttggtgta), and a TaqMan probe (5 µM) (5′-6FAM-ccgccgaactgagcagacacccgcgc-TAMRA, where 6FAM is 6-carboxyfluorescein and TAMRA is 6-carboxytetramethylrhodamine). The amplification was carried out using the Applied Biosystems 7300 Real-Time PCR System (Foster City, CA) under the following conditions: 10 min at 95 °C, 60 s at 95 °C for 40 cycles, 30 s at 60 °C, and 30 s at 72 °C. Absolute quantification Real-Time PCR using a specific TaqMan probe was performed to detect viral DNA. Viral load was derived from the threshold cycle (CT) using the standard curve generated in parallel, and the result is expressed as the concentration in µg of DNA/µL.

### 2.11. RNA Extraction and Real-Time PCR

Total RNA was extracted using TRIzol^®^ (Life Technologies, Carlsbad, CA, USA).), according to the manufacturer’s instructions, and DNase treatment before cDNA transcription was as follows: 1 μg of RNA was incubated at 37 °C for 2 h with 5 μL 10X DNase I Buffer, 2 μL recombinant RNase-free DNase I (10U) (2270A TaKaRa, Dalian, China), and RNase inhibitor (20U) (N251A Promega, Madison, Wisconsin, United States). The procedures were published previously in [28]. Total RNA (1 μg) was reverse-transcribed using ReverTra Ace^®^ qPCR RT Master Mix (FSQ-201 Toyobo, Kita-ku, Osaka, Japan) under the following conditions: 37 °C for 15 min, followed by 50 °C for 5 min and 98 °C for 5 min. The cDNAs were used for quantitative Real-Time PCR carried out using the Applied Biosystems 7300 Real-Time PCR System (Foster City, CA, USA). The thermal profile consisted of 10 min incubation at 95 °C followed by 30 cycles of 15 s denaturation at 95 °C, 35 s annealing at 60 °C, and 45 s elongation at 72 °C. The cDNA copy numbers were normalized to GAPDH. The analytic primers for RT-PCR were as follows: ICP0 forward 5′-TCTGCATCCCGTGCATGAAAAC-3′ and reverse 5′-CTGATTGCCCGTCCAGATAAAG—3′; UL42 forward 5′-CTCCCTCCTGAGCGTGTTTC-3′ and reverse 5′- CACAAAGCTCGTCAGTTCGC-3′; Us11 forward 5′- GGCTTCAGATGGCTTCGAG-3′ and reverse 5′- GGGCGACCCAGATGTTTAC-3′; GAPDH forward 5′-GAGAAGGCTGGGGCTCAT-3′ and reverse 5′- TGCTGATGATCTTGAGGCTG-3′. Each quantitative Real-Time PCR experiments included a minus-reverse transcriptase control.

### 2.12. Protein Extraction and Immunoblot Analysis

Immunoblot analysis was carried out to evaluate the accumulation of viral proteins. HeLa cells (4 × 105 cells/wells) and the virus were pre-treated with oleuropein (150 µg/mL, 300 µg/mL, and 400 µg/mL) for 1h at 37 °C. Then, the pre-treated virus was used to infect pre-treated cells. The infection was carried out at 10 MOI at 37 °C, and 1 h later, the viral inoculum was removed and replaced with growth medium in the presence of both compounds at different concentrations. Acyclovir (50 µM) was used as a positive control. Twenty-four hours post-infection, the samples were collected and lysed in SDS sample buffer 1X (62.5 mM Tris-HCl (Tris(hydroxymethyl)aminomethane hydrochloride) pH 6.8; 50 mM DTT (dithiothreitol); 10% glycerol; 2% SDS (sodium dodecyl sulfate); 0.01% bromophenol blue; and EDTA-free protease inhibitor cocktail 1X (Roche Basilea, Svizzera), sonicated, and boiled for 5 min. An equal amount of protein extract was subjected to SDS-gel electrophoresis (SDS-PAGE), transferred to membranes (Bio-Rad Life Science Research, Hercules, CA, USA), and subjected to immunoblot analysis. The procedure was published previously in [29]. Specific bands were visualized using Immobilon Classico Western HRP substrate (Merk, Millipore Darmstadt, Germany). Anti-GAPDH (sc-32233), anti-HSV-1 UL42 (sc-53333), and anti-ICP0 (sc-56985) were purchased from Santa Cruz Biotechnology (Santa Cruz, CA, USA). Phospho-c-Jun (Ser73) (D47G9) and phospho-c-Fos (Ser32) (D82C12) were purchased from Cell Signaling Technology (Beverly, MA, USA). Anti-PKR (phospho-T446, ab 32036) was purchased from Abcam (Cambridge, UK). Monoclonal anti-US11 was provided by Professor Bernard Roizman. Secondary HRP-conjugated goat anti-mouse IgG was purchased from Millipore. Quantitative densitometry analysis of immunoblot band intensities was performed using ImageJ software. The intensity of the target protein was divided by the intensity of the GAPDH and graphically represented using GraphPad Prism 6 software (GraphPad Software, San Diego, CA, USA).

### 2.13. Statistical Analysis

Three independent experiments were carried out in triplicate (n = 3) for each assay and the results represent the average ± standard deviation (SD). Statistical analysis was performed using GraphPad Prism 8 software (Graph-Pad Software, San Diego, CA, USA) using one-way variance analysis (ANOVA). The significance of the *p*-value is indicated with asterisks (*, **, ***, ****), which indicate significance of the *p*-value less than 0.1, 0.01, 0.001, and 0.0001, respectively. The half-maximal cytotoxic concentration (CC50) and the half-maximal effective concentration (EC50) values were calculated using non-linear regression analysis.

## 3. Results

### 3.1. Antioxidant Potential

The total antioxidant capacity of both OESA and OESY was determined according to the phosphomolybdenum assay [20,21] and the nitric oxide radical inhibition assay [22]. The procedure was described in the Material and Methods. The power of ammonium phosphomolybdate in the present study is expressed as the mg equivalent of vitamin C/g of dry matter. Our results indicate that OESA and OESY extracts show antioxidant activity equivalent to that of Vit C. (Table 1). Similarly, we found that the percentage of NO radical inhibition increased with an increase in the concentration of the extracts, with a half-maximal inhibitory concentration (IC50) of 0.15 and 0.17 mg/mL for OESA and OESY but was still less compared to the standard (Vit C) (IC50 = 0.058 mg/mL) (Table 1).

### 3.2. Cytotoxicity Effect of Olive Leaf Extracts

To investigate the cytotoxic effect of OESA and OESY on the HeLa human cell line, cells were treated with various concentrations of OESA and OESY (0.1; 0.2; 0.4; 1; 5; 10; and 20 mg/mL) for 72 h, and then, submitted to the cell viability assay (Figure 2). The data show that OESA and OESY affect cellular growth in a dose-dependent manner, and the IC50 was 2.66 mg/mL for OESA and 0.848 mg/mL for OESY. Based on the cell viability assay results, the 0.4 mg/mL and 0.2 mg/mL concentrations for both compounds, which induce less than 20% cytotoxicity, were chosen for further analysis.

### 3.3. Biological Antioxidant Activity in Human Cell Culture: MDA Assay

The MDA assay serves to monitor lipid peroxidation, which is a reaction to the oxidative degradation of polyunsaturated fatty acids mediated by oxygen-derived free radicals [23]. A final product of polyunsaturated fatty acid peroxidation in the cells during oxidative stress is MDA. To investigate the biological antioxidant activity of OESA and OESY, HeLa cells were cultured for 72 h and the levels of MDA were measured after the induction of oxidative stress by adding 0.2 mM H_2_O_2_ and after exposition to OESA or OESY for 30 min. Our data show a significant increase in the MDA adduct level after H_2_O_2_ treatment compared to the basal levels of MDA. A significant decrease in MDA levels was found following treatment with both extracts (Figure 3).

### 3.4. Antioxidant Enzyme Activities

Cells contain large numbers of antioxidants to prevent or repair the damage caused by reactive oxygen species. The bioactivity of OESA and OESY toward oxidative stress was measured by estimating the concentrations of enzymatic antioxidants such as catalase (CAT), superoxide dismutase (SOD), and glutathione peroxidase (GPX) in HeLa cells after the induction of oxidative stress. As shown in Table 2, the induction of oxidative stress with H_2_O_2_ led to a decrease in catalase, SOD, and GPx activities. Interestingly, cell treatment with OESA and OESY significantly increased catalase, SOD, and GPx activity (*p* < 0.001).

### 3.5. Antiviral Activity of OESA and OESY in HeLa Cells

To analyze the antiviral effect of OESA and OESY on HeLa cells, we measured the intracellular virus production following treatment with both extracts and infection with HSV-1 for 24 h. At 24 h post-infection (p.i.), intracellular infectious viral particles were released via freeze-and-thaw cycles, and the virion yields were determined via titration of the plaque-forming units (PFUs) in VERO cells (Figure 4). We found a strong reduction in viral titer following treatment with both extracts and compared with the untreated infected sample. These data suggest the significant incapability of HSV-1 to replicate on HeLa cells following OESA and OESY treatment. The half-maximal effective concentration (EC_50_) was calculated from concentration–effect curves using non-linear regression analysis. The EC_50_ values, CC_50_, and SI are shown in Table 3. Additionally, both extracts directly target the viral DNA, reducing the amount of viral DNA significantly (** *p* < 0.01) (Figure 5).

### 3.6. Antiviral Activity of OESA and OESY Compounds

The data reported in Figure 5 show that viral DNA accumulation is prevalently affected by OESA compared to OESY treatment. Thus, to understand whether OESA polyphenols content was responsible for the blockage of viral replication, we tested the antiviral activity of oleuropein. Our choice was supported by phytochemical characterization [16] in which oleuropein was among the most abundant compounds present in OESA and was completely absent in OESY. Before exploring antiviral activity, the viability assay was carried out by treating HeLa cells with oleuropein at different concentrations (Figure 6). The results show a good profile of tolerability to all concentrations of oleuropein (from 25 µg/mL to 400 µg/mL). Thus, non-toxic concentrations of oleuropein were employed to evaluate antiviral activity (Figure 7).

The results report that oleuropein drastically reduces the HSV-1 titer at 400 µg/mL and 300 µg/mL (Figure 7A). The half-maximal effective concentration (EC_50_) was calculated from concentration–effect curves using non-linear regression analysis. The EC_50_ values, CC_50_, and SI are shown in Table 4. Additionally, a significant reduction in viral transcripts, as well as proteins, of all three phases of the viral replication cascade was detected following treatment with 400 µg/mL and 300 µg/mL of oleuropein (Figure 7B,C, lane 2 vs. 4 and 5). The results suggest that oleuropein affects HSV-1 at different steps of viral replication.

### 3.7. Oleuropein Suppresses Viral Replication, Stimulating PKR-Modulated c-Fos and c-Jun Expression

The results reported above demonstrate that the treatment of HeLa cells with oleuropein, after viral adsorption, inhibits HSV-1 replication. As HSV-1 heavily relies on cellular signaling pathways for its effective replication, we hypothesized the involvement of the double-stranded RNA-activated protein kinase R (PKR) as a potential cellular response to viral replication. HSV makes a network of virus proteins that block this pathway, either directly or indirectly, but how these proteins work together is uncertain due to contradictory results [30]. Thus, the phosphorylation form of PKR in HSV-1 infected and oleuropein-treated HeLa cells was evaluated via Western blot analysis. Figure 8, panels A and B show that oleuropein treatment (400 µg/mL) in the infected cells increases the accumulation of phosphorylated PKR (p-PKR), as reported by the graphical representation of band intensity in panel B. Because PKR activation stimulates the inflammation-related pathway, including the c-Jun N-terminal kinase (JNK) pathway [31,32,33,34,35], the downstream PKR phosphorylation signaling cascade was evaluated. The expression of AP-1 family transcription factors, c-Fos, and c-Jun was also determined following oleuropein treatment during HSV-1 infection. The results report changes in p-PKR that correlated with the accumulation in the phosphorylated form of c-Fos (p-c-Fos) and c-Jun (p-c-Jun) (Figure 8C,D) in HSV-1-infected treated cells. Indeed, while HSV-infected cells result in an increase in the phosphorylated form of both c-Fos and c-Jun, compared to uninfected cells, oleuropein treatment at 400 µg/mL increases the accumulation of p-PKR, leading to a high phosphorylation of both c-Fos and c-Jun protein in a concentration-dependent manner. These results indicate that p-PKR accumulation, mediated by oleuropein treatment, allows for the activation of c-Jun and c-Fos and could be responsible for the antiviral mechanism employed by oleuropein to block viral replication.

## 4. Discussion

Some human physiological processes or pathological conditions, viral infection, and exposition to radiation, pollution, and cigarette smoke, lead to the generation of free radicals that, in physiological conditions, are neutralized by the endogenous antioxidant defense systems [36]. When these molecules gradually accumulate due to an imbalance between reactive oxygen species (ROS) generation and antioxidant defense systems, the free radicals trigger oxidative stress, which is responsible for irreversible changes in biomolecules [37]. Furthermore, free radicals can cause DNA mutations and interfere with the signal transduction mechanisms that regulate cell replication, differentiation, and death and are associated with various types of cancer development [38]. Similarly, cancer cells produce elevated levels of superoxide or H_2_O_2_, which can significantly aggravate the transformation of healthy cells into cancer cells. Thus, new strategies need to be developed to prevent and treat cancer and cancer related to viral infection. Polyphenols are a widespread group of secondary metabolites found in all plants which exhibit antioxidant activity, which can be useful in chemoprevention and chemotherapy. Several studies have demonstrated that antioxidants neutralize oxidative stress, either enzymatically or by increasing superoxide dismutase, catalase, or glutathione peroxidase. Thus, the modulation of oxidative stress by dietary agents could prevent or reduce cancer incidence [39]. The biological activities of crude extracts, bioactive component-enriched fractions, and pure compounds were demonstrated in in vitro and in vivo studies. In particular, they showed their application as antibacterial, antiviral, and anticarcinogenic agents [4,28]. The leaves of *Olea europaea* L. contain a higher quantity of phenolic compounds than that present in the fruit or in virgin olive oil [11,12,40,41], and olive leaf extracts have been confirmed to have a strong antiviral impact against various viruses [15,16,42,43]. In this study, we assessed the biological activity of two Mediterranean olives: the cultivated type (*O. europaea* L. ssp. *europaea* var. *sativa*) and the true wild oleaster (*O. e.* e. var. *sylvestris*), respectively reported as OESA and OESY. We demonstrated that OESA and OESY have strong free radical scavenging potential. The total antioxidant activity of both cultivars was determined via an ammonium phosphomolybdate assay and a nitric oxide radical inhibition assay. According to other studies [15,16,44], both extracts showed reducing abilities, expressed as mg/mL of Vitamin C, and the percentage of NO radical inhibition increased with an increase in the concentration of the extracts, with an IC50 of 0.15 and 0.17 mg/mL for OESA and OESY (Table 1). In addition, we reported that OESA and OESY did not exhibit an inhibitory effect on HeLa cell proliferation (IC50 of 2.66 mg/mL and 0.848 mg/mL, respectively). However, at higher concentrations only, they displayed an increase in the inhibitory effect on cell proliferation (Figure 2). Similarly, Zeriouh and coauthors showed that the phenolic extract from olive leaves exhibits a cytotoxic effect on the human colorectal cancer cell lines HCT116 and HCT8 in a dose-dependent manner [45]. Other studies reported that the cell growth and viability of MKN45 tumor cells are affected by a phenolic extract from *Olea europaea* olives in a dose- and time-dependent manner [46], as well as *Olea europaea* olive oil, which has a cytotoxic effect on two different cancer cell lines: T-47D and MCF-7 [47]. Additionally, once we assessed the antioxidant efficacy of both extracts, we estimated the lipid peroxidation induced by hydrogen peroxide in HeLa cells by measuring MDA, which is involved in cellular damage. In the present study, OESA and OESY significantly inhibited lipid peroxidation induced by hydrogen peroxide. This activity could be attributed to the cumulative antioxidant contribution of phenols and flavonoids. Oxidative treatment with 0.2 mM of H_2_O_2_ increased MDA levels due to enhancing the lipid peroxidation reaction. Contrarily, the pretreatment of cells with OESA and OESY reduced the production of MDA (Figure 3). Moreover, the simultaneous addition of H_2_O_2_ and extracts in the culture medium of HeLa cells increased catalase, superoxide dismutase, and glutathione peroxidase activities, suggesting an improvement in the adaptation of the enzymatic antioxidant system of the cells to ROS production/addition. In particular, the cultivated leaf extract treatment caused a higher increase in enzymatic activities than the wild leaf extract (Table 2). The significant increase in CAT activity may be due to the contribution of secoiridoids, which are a more abundant class of compounds identified in OESA (55.38%) compared to OESY (24.97%) [16]. Indeed, the comparison of the phytochemical profile of OESA and OESY showed that extracts are characterized by the presence of several compounds whose abundance could guarantee biological activity. Thus, treatment with OESA and OESY induced a significant increase in SOD and GPx enzymatic activities compared to the positive control in a concentration-dependent manner. Our results agree with previous works, which showed that the extract of *Olea europaea* exhibits a significant increase in antioxidant enzyme activity [48,49]. Thus, the MDA concentration of the pretreated cells was significantly lower (*p* < 0.01, *p* < 0.0001) and the activities of SOD, CAT, and GPx were significantly higher (*p* < 0.001) in the pretreated cells than in the positive control. These observations suggest that human cervical carcinoma HeLa cells could benefit from the antioxidant properties of the extracts during oxidative stress induction. Our previous study reported that OESA has a noticeable effect on the scavenging of free radicals and exhibits high antioxidant activity and a protective effect against lipid peroxidation during Epstein Barr virus lytic cycle induction [15]. Moreover, the antiviral activity of olive leaf extracts towards ILTV virus [50], VHSV virus [51], HIV-1 [52], and HSV-1 was previously reported [16,41]. Complementing these previous studies, we characterized the antiviral effects of OESA and OESY in a HeLa cancer model, demonstrating that treatment with both extracts significantly (*p* < 0.01) reduced the infectious viral particle production (Figure 4) affecting viral DNA synthesis (Figure 5), but the effect was greater for OESA than for OESY. Thus, to understand whether OESA polyphenol content was responsible for the blockage of viral replication, we tested the antiviral activity of oleuropein. Our choice was supported by phytochemical characterization [16] in which oleuropein was among the most abundant compounds present in OESA and was completely absent in OESY. Overall, our data demonstrate that oleuropein drastically reduces the HSV-1 titer at 400 µg/mL and 300 µg/mL (Figure 7A), viral gene expression in infected cells, as well as viral protein accumulation (Figure 7B,C). Oleuropein is known as an antioxidant, anti-inflammatory, and antiviral compound [53]. Here, we reported that the sub-toxic concentrations reduce viral replication more than acyclovir treatment (Figure 7C, lane 5 vs. 3, and Table 4). In addition, we discovered that oleuropein highly inhibits HSV-1 replication by activating the PKR-related host cell’s defense. As a component of the cellular antiviral response pathway, PKR is auto-phosphorylated and activated upon binding to the viral dsRNA of HSV-1 [30,54]. This allows for the inhibition of viral protein synthesis via the phosphorylation of eIF2α and triggers the transcription of inflammatory genes, including the c-Jun N-terminal kinase (JNK) pathway [31,32,33,34,35]. It is unclear if the multiple PKR inhibitory mechanisms are redundant roles or necessary functions to regulate PKR activity at different stages in the HSV-1 life cycle [30,55]. However, in our study, significant induction and accumulation of p-PKR was found following HSV-1 exposure, and this increased upon oleuropein treatment (Figure 8). Additionally, the coordinated accumulation of p-c-Fos and p-c-Jun proteins with p-PKR, observed in infected treated cells, suggests the activation of antiviral cellular signals which affect HSV-1 replication in HeLa cells. Data in the literature report that the dimerization of c-Jun and c-Fos enables antiproliferative signals in cancer cells and proinflammatory responses [34]. Thus, we hypothesize that oleuropein could protect the cells from viral replication, through the accumulation of the AP-1 cellular transcription factors that are related to the transactivation of proinflammatory genes.

## 5. Conclusions

The importance of this study arises from the ability of OESA and OESY polyphenols to have antioxidant activity and is correlated with the fact that a pure compound, such as oleuropein, activates antiviral cellular signals that affect HSV-1 replication in HeLa cancer cells. Although HSV-1 is not considered an oncogenic virus itself, it may increase the risk of malignant progression in vivo. Particularly, latent infection can be considered one feature necessary to contribute to oncogenesis. Indeed, after primary infection, the virus persists in the host and reactivates itself, altering the host immune system and the activity of cellular transcription factors, to support viral replication. The cancer cells are more susceptible to viral infections and thus the viruses take advantage of cellular processes to favor their life cycle creating a microenvironment favorable for activating uncontrolled cell proliferation. This scenario potentially causes the deterioration of cancer patients in comparison to non-cancer patients [56,57]. Targeting the virus represent a new therapeutic strategy against cancer development. Thus, one of the most beneficial aspects of our study was to individuate an approach to controlling HSV-1 replication with natural sources and downstream benefits, to prevent the accelerated development of tumors.

## Figures and Tables

**Figure 1 biomolecules-13-00238-f001:**
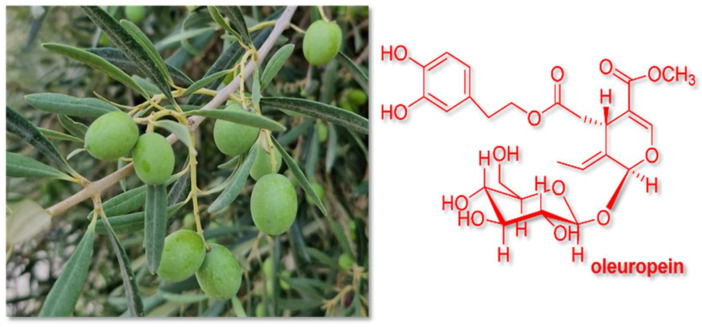
Photo of *Olea europaea* L. and chemical structure of oleuropein.

**Figure 2 biomolecules-13-00238-f002:**
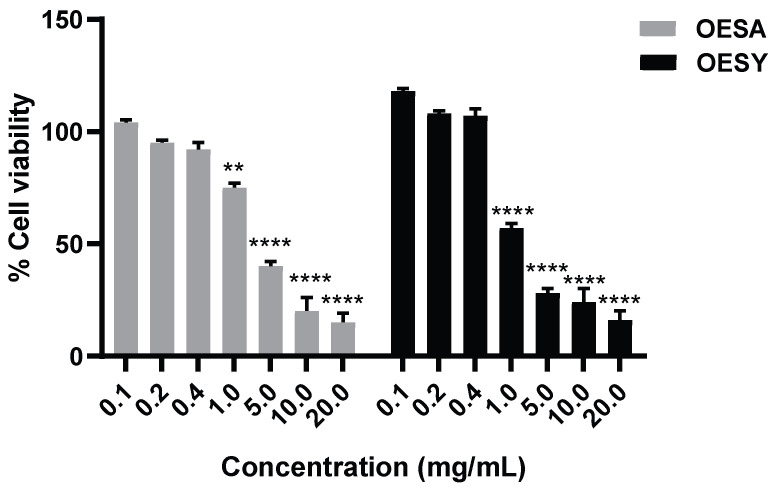
Cytotoxic effects of OESA and OESY on HeLa cell line. The inhibitory effects of OESA and OESY on cell growth were determined using a CCK8 assay. Cells were exposed to serial dilutions of OESA and OESY (0.1; 0.2; 0.4; 1; 5; 10; and 20 mg/mL) for 72 h and further incubated with CCK8 tetrazolium salt solution in the dark for 4 h. The absorbance was measured at 460 nm and the % of cell viability was calculated with respect to the untreated cells. Results are expressed as mean ± standard deviation (n = 3). ** *p* < 0.01, **** *p* < 0.0001.

**Figure 3 biomolecules-13-00238-f003:**
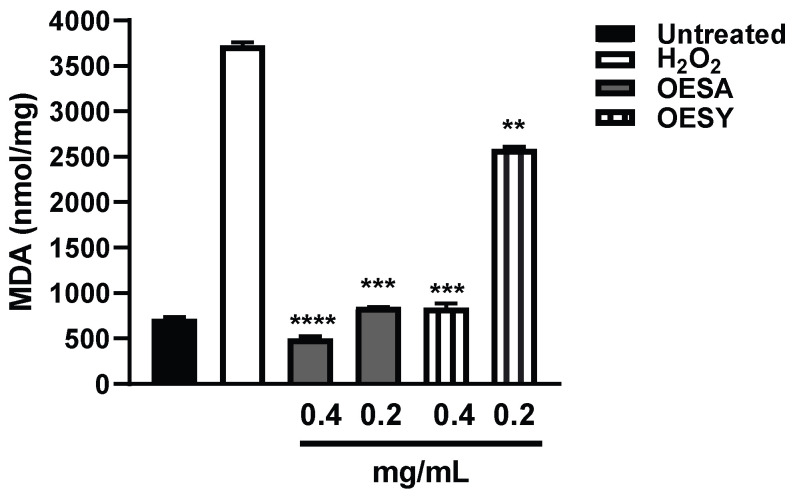
MDA levels in untreated HeLa cells and those treated with OESA and OESY. HeLa cells were cultured simultaneously and exposed to 0.4 and 0.2 mg/mL of OESA or OESY and H_2_O_2_ (0.2 mM) for 30 min. The MDA levels were evaluated by determining the thiobarbituric acid-reactive substances. The data are expressed in nmol/mg of protein (** *p* < 0.01, *** *p*< 0.001, **** *p* < 0.0001). Results are expressed as mean ± standard deviation (n = 3).

**Figure 4 biomolecules-13-00238-f004:**
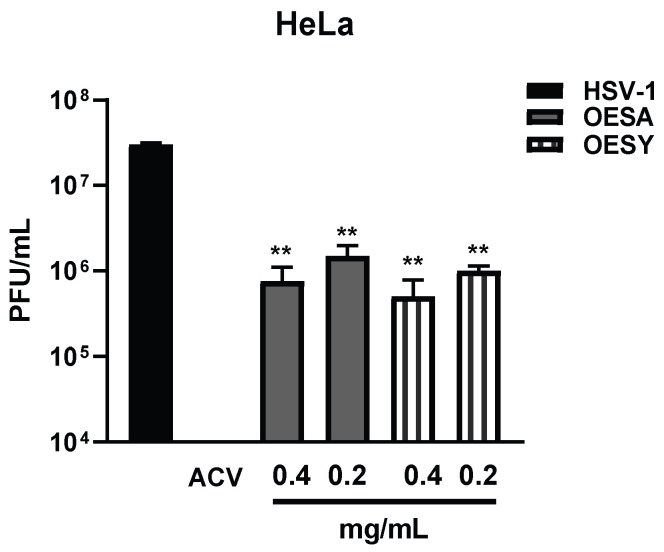
Evaluation of viral titer in HeLa cells following OESA and OESY treatment. HeLa cells (4 × 10^5^ cells/wells) and the virus were pre-treated with OESA and OESY at 0.4 and 0.2 mg/mL for 1 h at 37 °C. Then, the pre-treated virus was used to infect pre-treated cells. The infection was carried out at 10 MOI at 37 °C, and 1 h later, the viral inoculum was removed and replaced with growth medium in the presence of both compounds at 0.4 and 0.2 mg/mL. Acyclovir (50 µM) was used as a positive control. Samples collected 24 h p.i were frozen and thawed three times and 100 µL of each dilution of the viral suspension was used to infect the VERO monolayers. The multiwell plates were incubated for 1h at 37 °C. Then, the viral inoculum was removed and replaced with a culture medium containing 0.8% methylcellulose. After 72 h, plaques were visualized and counted on the microscope after staining with a crystal violet solution. Statistical analyses were performed in triplicate using a one-way ANOVA analysis assay, and ** *p* < 0.01 vs. HSV-1 +DMSO indicates significant change.

**Figure 5 biomolecules-13-00238-f005:**
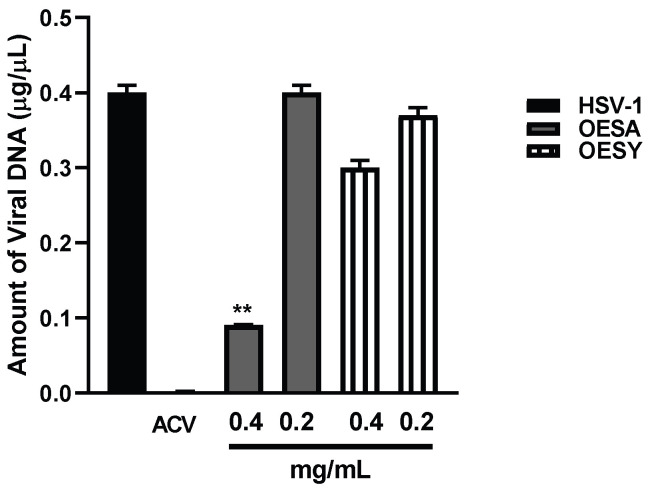
Absolute quantification of viral DNA. HeLa cells and the virus were pre-treated with OESA and OESY at 0.4 and 0.2 mg/mL for 1 h at 37 °C. Then, the pre-treated virus was used to infect pre-treated cells. The infection was carried out at 10 MOI at 37 °C, and 1 h later, the viral inoculum was removed and replaced with growth medium in the presence of both compounds at 0.4 and 0.2 mg/mL. Acyclovir (50 µM) was used as a positive control. The viral DNA was extracted 24 h post-HSV-1 infection as described in the Materials and Methods. Viral DNA amounts were determined via absolute Real-Time PCR using a TaqMan probe. The results are expressed as concentrations in µg of DNA/µL. Statistical analyses were performed in triplicate using a one-way ANOVA analysis assay, and ** *p* < 0.01 vs. HSV-1 + DMSO indicates significant change.

**Figure 6 biomolecules-13-00238-f006:**
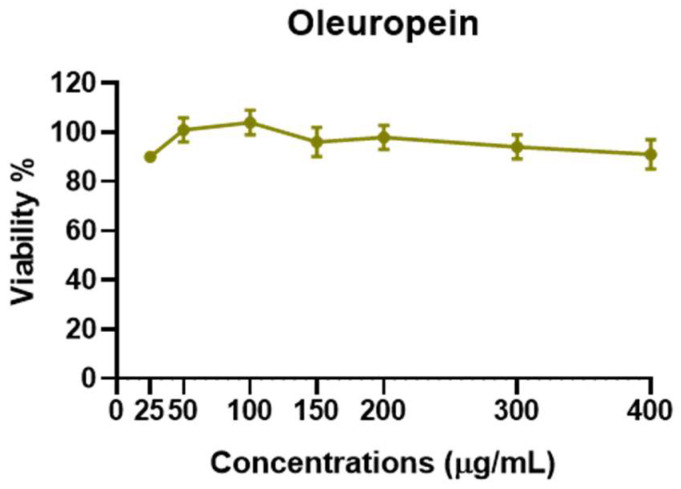
Cytotoxicity assay of oleuropein. HeLa cells were treated with different concentrations of oleuropein (25 µg/mL, 50 µg/mL, 100 µg/mL, 150 µg/mL, 200 µg/mL, 300 µg/mL, and 400 µg/mL). The cells were collected 72 h post-treatment, and the cell viability assay was carried out as described in the Section 2 and graphically reported as viability (%). The assay results are presented as the means of triplicates ± SD.

**Figure 7 biomolecules-13-00238-f007:**
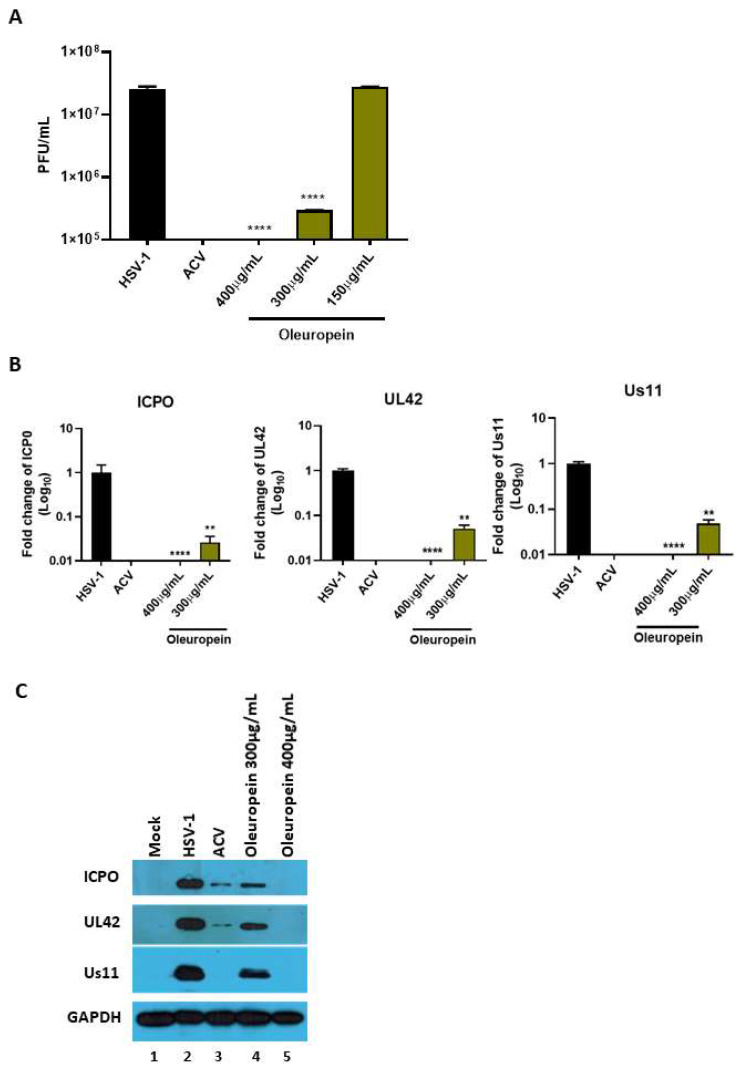
Antiviral activity of pure compounds. (**A**) HeLa cells (4 × 10^5^ cells/wells) and the virus were pre-treated with oleuropein (150 µg/mL, 300 µg/mL, and 400 µg/mL) for 1 h at 37 °C. Then, the pre-treated virus was used to infect pre-treated cells. The infection was carried out at 10 MOI at 37 °C, and 1 h later, the viral inoculum was removed and replaced with growth medium in the presence of both compounds at different concentrations. Acyclovir (50 µM) was used as a positive control. Virus yield was determined at 24 h p.i. using the standard plaque assay on VERO cells. (**B**) Relative quantification of viral transcripts (ICP0, UL42 and US11) was performed using Real-Time quantitative PCR and analyzed using the comparative Ct method (^ΔΔ^Ct). Values represent ± SD of the average of three independent experiments normalized against GAPDH. (**C**) Western blot analysis of ICP0, UL42, and Us11 viral proteins. GAPDH was used as a housekeeping gene. Data are expressed as a mean (± SD) of at least three experiments. Statistical analyses were performed using a one-way ANOVA analysis assay. ** *p* < 0.01, **** *p* < 0.0001 vs. HSV-1 + DMSO.

**Figure 8 biomolecules-13-00238-f008:**
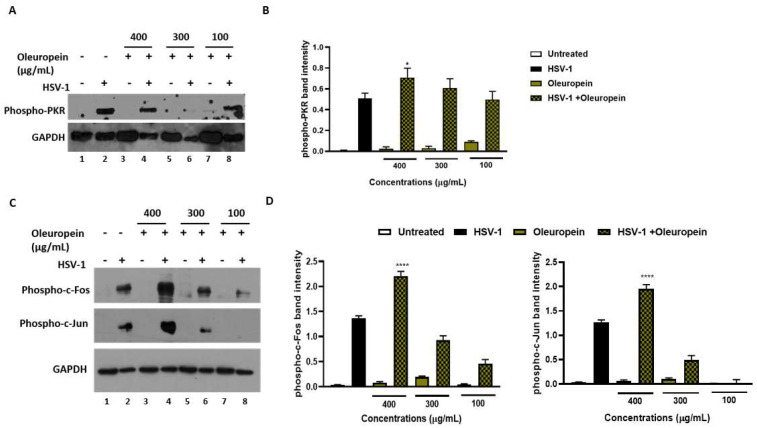
Activation of innate antiviral response mediated by phospho-PKR, phospho-c-Jun, and phospho-c-Fos in infected HeLa cells following oleuropein treatment. HeLa cells were infected at 10 MOI with HSV-1 and were untreated or treated with different concentrations of oleuropein (400 µg/mL, 300 µg/mL, and 100 µg/mL). (**A**–**C**) Phospho-PKR, phospho-c-Jun, and phospho-c-Fos protein expression was evaluated via Western blotting. Quantitation of the respective phospho-PKR, phospho-c-Fos and phospho-c-Jun protein bands is shown in (**B**,**D**). Data represent the means of 3 independent experiments ± SD. Statistical evaluation via one-way ANOVA: * *p* < 0.1, **** *p*  <  0.001.

**Table 1 biomolecules-13-00238-t001:** Antioxidant potency of OESA and OESY.

	OESA	OESY	Vitamin C
TAA ^1^ (mg/mL)	0.8 ± 0.2	3.4 ± 0.5	1
NO ^2^ (mg/mL)	0.15 ± 0.01	0.17 ± 0.02	0.058 ± 0.03

^1^ TAA: total antioxidant activity expressed as EC50 (mg/mL) value calculated by ammonium phosphomolybdate assay; OESA: *O. europaea* var. *sativa*; OESY: *O*. *europaea* var. *sylvestris.* ^2^ NO: anti-radical activity of OESA and OESY extracts against NO radical. The results are expressed as calculated IC50 (mg/mL) value.

**Table 2 biomolecules-13-00238-t002:** Effects of pretreatment with OESA and OESY on CAT, SOD, and GPx activity.

Treatment	Untreated	H_2_O_2_	OESA (0.4mg/mL)	OESA (0.2mg/mL)	OESY (0.4mg/mL)	OESY (0.2mg/mL)
SOD ^1^ (U/mg protein)	41.78 ± 1.22	32.35 ± 2.94	41.45 ± 1.76 ***	38.20 ± 1.47 ***	39.32 ± 1.41	29.90 ± 5.97
CAT ^2^µmol H_2_O_2_/mg protein	34.26 ± 0.80	19.03 ± 1.46	32.84 ± 1.20 ***	29.52 ± 2.57 ***	30.99 ± 2.6 ***	14.88 ± 2.9 ***
GPx ^3^ µmol GSH/min/mg protein	0.93 ± 0.001	0.19 ± 0.009	0.30 ± 0.01 ***	0.20 ± 0.02 ***	0.35 ± 0.006 ***	0.24 ± 0.008 ***

^1^ SOD: superoxide dismutase; ^2^ CAT: catalase activity; ^3^ GPx: glutathione peroxidase; OESA: *O. europaea* var. *sativa*; OESY: *O. europaea* var. *sylvestris*. *** *p* < 0.001.

**Table 3 biomolecules-13-00238-t003:** Selectivity index (SI), cytotoxicity (CC_50_), and antiviral activity (EC_50_) of OESA and OESY.

Extract	CC_50_ (mg/mL)	EC_50_ (mg/mL)	SI
OESA	2.66	0.15	17.7
OESY	0.848	0.17	4.98

CC_50_: half-maximal cytotoxic concentration; EC_50_: half-maximal effective concentration; SI: selectivity index, the ratio of EC_50_/CC_50_. OESA: *O. europaea* var. *sativa*; OESY: *O. europaea* var. *sylvestris*.

**Table 4 biomolecules-13-00238-t004:** Selectivity index (SI), cytotoxicity (CC_50_), and antiviral activity (EC_50_) of oleuropein.

Extract	CC_50_ (mg/mL)	EC_50_ (mg/mL)	SI
Oleuropein	1.7	0.241	7

CC_50_: half-maximal cytotoxic concentration; EC_50_: half-maximal effective concentration; SI: selectivity index, the ratio of EC_50_/CC_50_.

## Data Availability

The data presented in this study are available on request from the corresponding author.

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
