# Peer review of "Analysis of Antioxidant and Antiviral Effects of Olive (Olea europaea L.) Leaf Extracts and Pure Compound Using Cancer Cell Model"

_biomolecules, 2023, doi:10.3390/biom13020238_

Round 1

Reviewer 1 Report

The manuscript entitled “Analysis of Antioxidant and Antiviral Effect of Olive (Olea europaea L.) leaf Extracts and Pure Compounds Using Cancer Cells Model” presents the evaluation of the antioxidant and antiviral potential of two hydroalcoholic extracts from Olea europaea L. var. sativa and Olea europaea L. var. sylvestris leaves.

The manuscript is quite interesting and well-written. However, major revisions should be made in order to be published in Biomolecules journal, and the manuscript should be completed and/or modified taking into account the suggestions from the attached file.

Author Response

Response to Reviewer 1:

We sincerely thank the reviewer for the comments which were of great help in revising the manuscript. Accordingly, the revised manuscript has been systematically improved. Below, you will find a point-by-point description which includes the original reviewer comments in boldface and the responses in red regular typeface. We specify that we resubmit the new version of the manuscript in which the number of the figures does not correspond to the previous version.

Reviewer 2 Report

Dear Authors.

The use of compounds of natural origin in the prevention of diseases is still an important research direction in modern science. Modern man is turning to nature in his search for new methods of disease prevention and treatment. The research performed here is part of this trend and accurately depicts the potential of the natural compounds studied. My doubts are raised by the use of a tumour line in the anti-free radical assessment, for this type of research it is better to use normal cells. 

1 Why was the HeLa line chosen for the study and not the Vero.

2. No description of the Vero line in the Materials and methods section.

3. The bar graphs are not very clear, too small, the border of the bars makes the standard deviation not readable.

4. In my opinion, the conclusions of the paper should be corrected, so that it does not follow from them that the obtained compounds protect cancer cells. HeLa was the model. 

5. The anti-free radical activity is chemopreventive. This should be addressed in the introduction and discussion. 

Author Response

Response to Reviewer 2:

We sincerely thank the reviewer for comments which were of great help in revising the manuscript. Accordingly, the revised manuscript has been systematically improved. Below, you will find a point by point description which includes the original reviewer comments in boldface and the responses in red regular typeface. We specify that we resubmit the new version of the manuscript in which the number of the figures not corresponds to previous version.

Round 2

Reviewer 1 Report

Although the authors made several changes and the manuscript has been significantly improved, I cannot agree with the response given by authors at no. 7 in coverletter. In botanical research, a voucher number is very important, and it should be given. As stated in several research papers, the voucher serves as the supporting material for published studies of the taxon and ensures that the science is repeatable, as a preserved specimen of an identified taxon deposited in a permanent and accessible storage facility. Vouchers are crucial in authenticating the taxonomy of an organism, as a tool for identifying localities of the taxon, and for additional taxonomic, genetic, ecological, and/or environmental research.  Many journals focused on the plant sciences require submission of vouchers as a condition for publication of articles.

Author Response

Dear Reviewer, we are grateful for your comment, below we report the information you have requested based on the Editor decision. We hope they are sufficient to give correct information based on the Biomolecules rules. We insert the additional informations in the section 2.1.

2.1 Plant materials and extraction procedure
"The samples of the olive leaves extracted from O. europaea var. sativa (OESA) and O. europaea var. sylvestris (OESY) were produced from the variety Chemlali SFAX. Olive leaves were collected manually from the National agricultural complex “Chaal” located in the Sfax region (Latitude: 34°44’26’’ North; Longitude: 10°45’37’’ East), Tunisia.  Samples were harvested in the same pedoclimatic conditions during the month of October 2018." 

Reviewer 2 Report

Dear authors.

Thank you for your very comprehensive and detailed answers to my questions and the corrections posted in the paper. I do not comment on the paper and recommend it for publication. 

Author Response

Based on the Editor decision we insert the additional informations in the section 2.1.

2.1 Plant materials and extraction procedure
"The samples of the olive leaves extracted from O. europaea var. sativa (OESA) and O. europaea var. sylvestris (OESY) were produced from the variety Chemlali SFAX. Olive leaves were collected manually from the National agricultural complex “Chaal” located in the Sfax region (Latitude: 34°44’26’’ North; Longitude: 10°45’37’’ East), Tunisia.  Samples were harvested in the same pedoclimatic conditions during the month of October 2018."